# Consumption of a Gelatin Supplemented with the Probiotic Strain *Limosilactobacillus fermentum* UCO-979C Prevents *Helicobacter pylori* Infection in a Young Adult Population Achieved

**DOI:** 10.3390/foods11121668

**Published:** 2022-06-07

**Authors:** Cristian Parra-Sepúlveda, Kimberly Sánchez-Alonzo, Joaquín Olivares-Muñoz, Cristian Gutiérrez-Zamorano, Carlos T. Smith, Romina I. Carvajal, Katia Sáez-Carrillo, Carlos González, Apolinaria García-Cancino

**Affiliations:** 1Department of Microbiology, Faculty of Biological Sciences, Universidad de Concepción, Concepción 4070386, Chile; cparras@udec.cl (C.P.-S.); kimsanchez@udec.cl (K.S.-A.); jolivares2018@udec.cl (J.O.-M.); cgutierrezz@udec.cl (C.G.-Z.); csmith@udec.cl (C.T.S.); rominacarvajal@udec.cl (R.I.C.); carlosg@udec.cl (C.G.); 2Department of Statistics, Faculty of Physical and Mathematical Sciences, Universidad de Concepción, Concepción 4070386, Chile; ksaez@udec.cl

**Keywords:** *Helicobacter pylori*, probiotic, *Limosilactobacillus fermentum*, anti-*H. pylori* activity, prevention

## Abstract

*Helicobacter pylori* is a bacterium associated with various gastrointestinal diseases of high worldwide prevalence. Since probiotics are an emerging alternative to managing infection by this pathogenic bacterium, the present work evaluated, in a randomized double-blind study controlled by a placebo, if consuming *Limosilactobacillus* *fermentum* UCO-979C prevents *H. pylori* infection in humans. Participants consumed either *L. fermentum* UCO-979C-supplemented gelatin (67 participants) or placebo-supplemented gelatin (64 participants) once a day, five days per week for 12 weeks. *H. pylori* infection in the participants was controlled before and after the intervention detecting *H. pylori* antigens in stools. Regarding *H. pylori*-infected participants before the study, 100% remained infected at the end of the study in the placebo group, while 96.7% of those receiving the probiotic remained infected after the intervention. Most importantly, of the non-infected participants, 34.2% became infected and 65.8% remained non-infected in the placebo group, while 2.7% became infected and 97.3% remained as non-infected individuals in the intervened group. Therefore, consuming the *L. fermentum* UCO-979C strain significantly reduced *H. pylori* infection, demonstrating a 92.6% efficacy in avoiding infection by this pathogen in non-infected individuals; thus, this probiotic is an excellent candidate to prevent *H. pylori* infections in non-infected individuals.

## 1. Introduction

*Helicobacter pylori* is a Gram-negative bacterium with the ability to colonize the stomach wall and other structures of the host [1]. This infection is associated with the development of clinical illnesses, such as chronic gastritis, peptic ulcers, mucosa-associated lymphoid tissue (MALT) lymphoma and gastric cancer [2]. It is estimated that infection by this pathogen affects nearly 50% of the global population [3,4]. Nevertheless, its prevalence varies according to geographical areas, it being higher in less developed countries [3]. Several studies have reported that the prevalence of *H. pylori* infection in Chile reaches approximately 73% in adults, and that it varies between 18% and 34% in pediatric patients [4,5,6]. Therefore, effective therapy is required to contribute to reducing these figures worldwide. For a long time, the first line of action against this pathogen has been triple therapy, including a proton pump inhibitor (PPI), clarithromycin and amoxicillin or metronidazole. Nevertheless, the effectiveness of this treatment has been markedly reduced during the last decade, with variations depending on the reported antibiotic resistance of the pathogen in different geographical regions [7]. The increased resistance of *H. pylori* to antimicrobials and the secondary effects associated with the antibiotics used to eliminate infection by this pathogen are the most important factors explaining the reduction in time of the efficacy of the anti-*H. pylori* treatments [8]. Therefore, it is necessary to search for innovative strategies to both treat and/or prevent infection by *H. pylori,* such as probiotics, live microorganisms which when administered in adequate concentrations confer a health benefit to the host based on their already reported benefits and solid evidence and other possible candidates to be used as anti-*H. pylori* alternatives [9].

However, to be probiotics, bacterial strains must be supported by clinical studies demonstrating their beneficial effects on the well-being of the hosts [10]. Diverse beneficial attributes of probiotics have been reported, including those having the potential to be used in clinical practice, such as nutritional probiotics, probiotic medications and probiotic foods of medical or common use [10]. Functional foods, mainly those including probiotics in their composition, are nowadays relevant due to their advantages which, associated with the permanent search to improve quality of life, prompt consumers to increase the demand for foods benefiting their organisms [11,12]. Presently, the market offers a large variety of dairy matrix-based functional foods with a diverse probiotic content [13,14]. Nevertheless, they might represent a disadvantage for persons unable to consume lactose. For this reason, in the present work, we evaluated a different type of matrix, gelatin, which has the advantage of lacking lactose, thus, extending the benefit of consuming a probiotic to a larger population. Gelatin is highly valued in the food industry due to its nutritious properties, its nearly 90% content of proteins, mainly collagen, absence of cholesterol and its functional properties, including water trapping [15]. Moreover, its low cost is within the reach of most of the population [16]. Considering its technological and functional properties, gelatin is extensively used in the industry to produce edible coatings which provide protection and stability to foods [17]. Moreover, it has been demonstrated that the encapsulation of probiotic bacteria in gelatin matrices improves their viability and protects them, improving the stability of probiotics in foods [18,19,20].

Our research group conducted a detailed characterization of the probiotic properties and the anti-*H. pylori* activity of the *Limosilactobacillus fermentum* UCO-979C strain, previously known as *Lactobacillus fermentum* UCO-979C, a lactic acid bacterium isolated from the human stomach [21,22], demonstrating that this strain is capable of tolerating an acidic pH and bile salts, producing hydrogen peroxide, and that it is highly hydrophobic [23,24]. Moreover, it was observed that this strain is able to efficiently adhere to the gastric mucosa, as shown by its in vitro adherence to gastric carcinoma human cells (AGS) and its in vivo adherence to the gastric mucosa in the Mongolian gerbil model [25]. Furthermore, it was demonstrated that *L. fermentum* UCO-979C strongly inhibits the adhesion, growth and the urease activity of *H. pylori* and that it improves resistance against infection by this pathogen, modulating the gastric innate immune response [26]. Previous results in humans also indicated that consuming the strain *L. fermentum* UCO-979C significantly improved healthy feeding habits and also increased the lean mass in the group consuming this probiotic [27]. Therefore, the present study corresponds to a final experimental stage to demonstrate both the effectivity and innocuity of the probiotic strain *L. fermentum* UCO-979C. Hence, the aim of this study was to evaluate the impact of administering gelatin supplemented with the probiotic strain *L. fermentum* UCO-979C on the *H. pylori* infection in asymptomatic Chilean young adults.

## 2. Materials and Methods

### 2.1. Design and Supervision

This research was a randomized double-blind study controlled by placebo to evaluate a treatment against *H. pylori* by administering the probiotic strain *L. fermentum* UCO-979C in gelatin. This study was approved and assigned the code CEBB 225-2018 by the Committee of Ethics, Bioethics and Biosafety, Vice-rectory of Research and Development, University of Concepcion, Concepcion, Chile. This assay was carried out in accordance with the indications of the Declaration of Helsinki [28].

The authors attest to the accuracy and integrity of the data collected and the strict observance of the protocol while performing the present study. This study was financed by the INNOVA call, Chilean Economic Development Agency (CORFO), Chile, and funds of the Laboratory of Bacterial Pathogenicity, Faculty of Biological Sciences, University of Concepción, Concepcion, Chile. The gelatin biomass and the placebo were supplied by the Laboratory of Bioprocesses, Department of Chemical Engineering, Faculty of Engineering, University of Concepcion, but it had no other participation in this study. The researchers had full autonomy to design and carry out this study and to prepare the manuscript.

### 2.2. Eligibility and Recruiting of Participants

One hundred and thirty-one University of Concepción (Chile) students were recruited to participate in this study. Invitations to participate were sent by e-mail or communicated at lecture halls. Invitations were issued between March and May 2018. The criteria for eligibility included being a University of Concepción (Concepción, Chile) student not consuming probiotics, prebiotics or medications for one month prior to the beginning of the study. Selection criteria also included being between 18 and 30 years old because this was the age group showing a progressive increase in *H. pylori* infection [5]. Criteria for exclusion from the study included pathological conditions, pregnancy, breastfeeding or severe depression. Students planning to become pregnant during the first 8 weeks of the study or considering surgery or other medical intervention were also excluded from the study. Qualified members of our research team explained the study to each candidate participating in this study and obtained written informed consent from each one of them.

### 2.3. Procedures 

It was determined that this study required at least 131 participants. This size of samples considered up to a 10% desertion of participants and 10% of eliminated participants due to their unfulfillment of the rules. As shown in Figure 1, 131 participants (79 women and 52 men) were randomly divided into two groups. The intervened group included 67 participants that received the probiotic strain *L. fermentum* UCO-979C in the gelatin, while the control group included 64 participants that received the same gelatin in which the probiotic was replaced by a placebo. Each participant received 5 servings of gelatin per week, and they were instructed to keep them refrigerated.

### 2.4. Culture of the Probiotic Bacterial Strain, Production of the Probiotic Biomass and Preparation of Gelatin 

*L. fermentum* UCO-979C was a bacterial strain obtained from a human gastric biopsy and was maintained at the culture collection of the Laboratory of Bacterial Pathogenicity, Department of Microbiology, University of Concepcion, Chile. The strain was cultured in Mann–Rogosa–Sharpe (MRS) broth (Difco, Wokingham, UK) under microaerobic conditions at 37 °C for 24 h. Then, an inoculum was transferred to MRS broth (BD Difco, Sparks, MD, USA) and incubated under the same atmospheric conditions for a further 24 to 48 h. The probiotic biomass of *L. fermentum* UCO-979C required to formulate the gelatin was prepared in a culture medium (patent number 1940–2005, Instituto Nacional de Propiedad Industrial (INAPI), Santiago, Chile) containing whey cheese, lactase, casein peptone and yeast extract with a pH of 6.8. The culture was kept under aerobic conditions at 37 °C in a 15 L fermenter. The biomass obtained was resuspended in 20% skim milk, frozen at −45 °C, lyophilized using a model BK-FD10PT freeze dryer (Biobase, Shandong, China) and kept at 4 °C in the dark until used. Counts indicated the presence of 10^10^ CFU g^−1^ of lyophilized powder [29]. Regarding the preparation of the gelatins, the probiotic strain *L. fermentum* UCO-979C, >10^7^ colony-forming units (CFUs) mL^−1^, was incorporated into three-layer gelatin (100 g serving) (Figure 2). The top and bottom layers consisted of flavored gelatin without saccharose, while the middle layer, containing the probiotic, was prepared with unflavored gelatin without saccharose. The placebo gelatin was identical to the probiotic gelatin in flavor and texture, except that it did not contain the probiotic strain, which was replaced by 2% skim milk. Gelatins were prepared weekly and random samples were analyzed by plate count to confirm that the probiotic strain counts were >10^7^ CFU mL^−1^. Plate counts were performed on MRS agar (Difco, Wokingham, UK) under microaerobic conditions at 37 °C for 24 h.

### 2.5. Data Collection

The demographic and clinical characteristics of the participants were evaluated before starting and after the end of the 12 weeks of the study. Age in years, gender (woman, man) and pathologies such as irritable bowel, diabetes, lupus, gastroesophageal reflux, gastritis, lactose intolerance, alcohol consumption, asthma and allergies were evaluated. Each participant was requested, before starting and after the end of the study, to provide a fecal sample to detect *H. pylori* using the Premier Platinum HpSA Plus Kit (Meridian Bioscience Europe, Italy), which detects *H. pylori* antigens in stools. This assay allowed the detection of *H. pylori-infected* participants before starting the assay to evaluate the effect of consuming the probiotic gelatin for 12 weeks on *H. pylori*-infected and non-infected individuals.

The primary efficacy of probiotic gelatin or placebo was concluded, evaluating the manifestations of adverse effects during the intervention period. An adverse event was defined as any event or reaction, independently of its causality, requiring hospitalization or considered as an important medical event, such as taste disturbance, diarrhea, abdominal pain, constipation, abdominal distension, nausea or vomiting. Basal demographic data of the participants were also collected and the participants were considered to comply with the instructions when they consumed more than 80% of the gelatins they received. Data were evaluated weekly by means of telephone interviews, performed by qualified researchers, or by means of questionnaires uploaded to a database. The unmasking of the assay was performed at the end of the analysis of the data.

To obtain the secondary efficiency results of the intervention in the participants consuming the probiotic gelatin or the placebo, the difference in the risk to acquire the *H. pylori* infection was calculated. It was calculated as indicated by Doll and Hill [30]:Risk among non-intervened group-Risk among intervened group
Risk among non-intervened group

### 2.6. Statistical Analysis 

The statistical analysis was performed using the SPSS software version 24.0 (IBM Company, Armonk, NY, USA). Analyses were conducted independently. Results were considered as statistically significant when the value of *p* was <0.05.

## 3. Results

### 3.1. Recruiting and Participants

A total of 182 students enrolled to participate in the study. From this total, 51 were excluded because they did not fulfill all the required criteria to be included. The remaining 131 students selected to participate were distributed by simple randomization into two groups, one receiving the probiotic-supplemented gelatin and the other receiving the placebo gelatin. Among the 131 recruited participants, 67 received the probiotic and 64 received the placebo. None of the 131 participants were eliminated from the study due to the loss of tracking or nonfulfillment problems during the study. Figure 1 shows the CONSORT flow diagram of recruitment and the analysis of the participants.

### 3.2. Evaluation of Participants

Table 1 shows the demographic and clinical characteristics of the participants recruited for the study. Age, alcohol consumption and comorbidities were similar between the two groups. There was no significant difference in the age of the participants. With respect to gender, women were dominant in both groups. Regarding diseases, no significant differences were detected when both groups were compared. During the 12 weeks of the intervention period, none of the participants reported adverse effects, suffered an infection or required the administration of antibiotics.

### 3.3. Viability of the Probiotic in the Gelatin

The counts of the probiotic in random servings of probiotic gelatin demonstrated that the concentration of *L. fermentum* UCO-979C was always above >10^7^ CFU mL^−1^ up to the third week after preparation of the gelatin. After four weeks, the concentration decreased by one logarithm (data not shown). It must be taken into consideration that the gelatins were prepared and distributed weekly; therefore, participants consumed the here reported concentrations of the probiotic.

### 3.4. Detection of H. pylori Using an Immunoassay in Stools before and after Consuming the Probiotic

Table 2 shows that there were no significant differences (*p* > 0.9999) between the two groups when comparing the number of *H. pylori*-infected participants before and after the probiotic treatment. The infected participants of the placebo group remained infected until the end of the assay (26/26), while from the participants of the intervened group, 29/30 remained infected post-intervention and 1/30 showed no *H. pylori* infection. On the other hand, there was a significant difference (*p* = 0.0005) when the participants not infected with *H. pylori* were analyzed before and after the intervention. Among the participants of the placebo group, 13/38 became infected during the time span of the assay and 25/38 remained as non-infected, while in the group which consumed the gelatin supplemented with probiotic, only 1/37 participants became infected and 36/37 remained as non-infected. This information allowed us to determine that the prevalence of *H. pylori* infection was 42.7%. Furthermore, according to the results, the estimated efficacy of the probiotic *L. fermentum* UCO-979C to prevent *H. pylori* infection was 92.6% in Chilean young adults (18 to 30 years old).

Figure 3A shows that, in accordance with the results of the detection of *H. pylori* antigens in their stools, the participants infected before the beginning of the intervention who received the placebo remained infected by the pathogenic bacterium *H. pylori* in 100% of the cases. With respect to the group receiving the probiotic gelatin, the infection persisted in most of the participants (96.7%) and only in 3.3% was the infection no longer present after the intervention. Figure 3B represents the *H. pylori* non-infected participants before consuming the probiotic gelatin or placebo, showing that among the participants who consumed the placebo, 34.2% became infected, while 65.8% remained non-infected, and of the participants receiving the probiotic, 2.7% became infected, while 97.3% remained free of the infection.

All the 131 participants consumed more than 80% of the probiotic or placebo gelatin scheduled doses. None of the participants in this study showed any adverse effect or important medical events. 

## 4. Discussion

Between 15% and 20% of the individuals infected with *H. pylori* develop a benign or malignant gastroduodenal pathology associated with the chronic inflammation caused by this bacterium [31]. These pathologies include, among others, peptic ulcer, MALT lymphoma and gastric cancer. The high prevalence of *H. pylori* at young ages has been associated with a high gastric cancer mortality rate in the Chilean population [5].

The eradication of *H. pylori* is associated with the healing of ulcers, regression of MALT lymphoma and a reduced risk of cancer [32]. Therefore, the importance of the eradication of this bacterium cannot be overemphasized [33]. Presently, the antibiotic treatment of infection by *H. pylori* is mandatory to prevent severe diseases; nevertheless, failures of the eradication therapy, particularly in the case of clarithromycin, have been observed worldwide [34,35]. In this respect, a study by our group indicated that clinical *H. pylori* isolates obtained in the Bio Bio region (Chile) from 2015 to 2017 were still highly sensitive to amoxicillin and tetracycline, but the resistance frequencies to clarithromycin (29.2%) and levofloxacin (20.8%) [36] suggested that the conventional triple therapy and therapeutic alternatives may not be effective in the Bio Bio region. A number of therapeutical alternatives have been investigated to prevent or to treat *H. pylori* infection [37]. Although vaccination must be highlighted as a preventive alternative, thus far, there is no vaccine available at the worldwide level to prevent infections by this bacterium. Considering the diversity of the immune response of individuals and the characteristics of *H. pylori* infection (large genetic heterogeneity of strains), there is still a long way to obtain an effective vaccine [38,39].

Probiotics, live microorganisms which when consumed in appropriate quantities confer beneficial effects on the health of the host beyond their primary nutritional effect [40,41], have been used to treat a number of diseases, such as inflammatory bowel disease, irritable bowel syndrome and diarrhea [42,43]. Aiba and coworkers demonstrated, for the first time, that *Lactobacillus acidophilus* inhibited the growth of *H. pylori* in vitro [44]. Furthermore, they indicated that the mechanism could be related to reduced urease activity mediated by the short-chain fatty acids produced by the probiotic, an improvement of the acidic environment of the stomach, damage to the cell wall of *H. pylori* or the inhibition of the colonization of the gastric mucosa by *H. pylori* [45,46,47].

Considering the antibiotic resistance shown by numerous bacteria, including *H. pylori*, probiotics appear as a very good preventive alternative [48]. The strain *L. fermentum* UCO-979C, isolated and characterized by our group, showed to possess good probiotic functional properties and to be innocuous [24,49,50]. Considering its characteristics and beneficial properties, this strain was used to prepare a probiotic gelatin, which was administered to young Chilean university students. This population was selected based on a study performed in Chile by Ferreccio and coworkers [5], who reported that gastric adenocarcinoma and a high *H. pylori* prevalence at early ages were some of the main causes of cancer-related mortalities in the country.

Carrion et al. [21] reported no adverse effects in the first human study using the strain *L. fermentum* UCO-979C. Our study showed that the probiotic strain *L. fermentum* UCO-979C consumed for 12 weeks was safe and that it did not produce diarrhea or gastrointestinal malaise. Therefore, this was the second local study providing evidence that the *L. fermentum* UCO-979C strain is safe to be consumed by humans. The present study, a double-blind assay controlled by placebo, demonstrated that the strain *L. fermentum* UCO-979C provided a statistically significant (*p* = 0.0005) prophylactic effect, preventing infection with *H. pylori* in those participants which at the beginning of the study were free of infection with this bacterium and consumed the probiotic gelatin for 12 weeks. Random gelatin servings were used to evaluate, once a week for four weeks, the number of viable CFU mL^−1^ of the probiotic in the gelatin matrix. The results demonstrated that the concentration of the *L. fermentum* UCO-979C strain was >10^7^ CFU mL^−1^ for three weeks, and that it decreased by one logarithm in the fourth week. Since the gelatin servings were prepared and distributed on a weekly basis, the participants of the intervened group always received gelatins containing >10^7^ viable CFU mL^−1^. Some studies have shown a direct relationship between the addition of potential probiotic strains and the in vitro inhibition of *H. pylori* growth [51]. The *L. acidophilus* [52] and *Lactobacillus casei* Shirota strains [53] have an antagonistic effect on *H. pylori*. Several reports describe probiotic strains with anti-*H. pylori* activity being mediated by the production of antimicrobial compounds, such as bacteriocins, autolysins and organic acids [54,55]. With respect to the present work, the prophylactic effect on infection by *H. pylori* could be due to the production of lactic acid by the *L. fermentum* UCO-979C strain [50], which reduces the pH and, therefore, reduces the activity of the *H. pylori* urease enzyme, causing an antimicrobial effect [50]. This effect has also been reported in *Lactobacillus salivarius*, *L. acidophilus*, *Lactobacillus rhamnosus* and *L. casei* Shirota strains, both in in vitro and in animal studies [56].

The prophylactic effect described in the present study seems to be specific for this particular strain [52,57,58]. Since *L. fermentum* UCO-979C is a good biofilm-forming bacterium, it may help to protect the gastric epithelium from the colonization of *H. pylori* [21]. The prophylactic effect of *L. fermentum* UCO-979C was already described by Merino et al. in an in vivo model [25], demonstrating that Mongolian gerbils which previously received the probiotic *L. fermentum* UCO-979C strain showed reduced colonization by *H. pylori* in the stomach antrum (87% reduction, *p* = 0.004) and stomach body (77% reduction, *p* = 0.0476). To the best of our knowledge, the prophylactic activity of the strain *L. fermentum* UCO-979C against *H. pylori* has been also previously reported for other lactobacilli [52,58].

The participants who, based on the detection of *H. pylori* in their stools, showed to be positive for the pathogen at the beginning of the study, showed no significant differences at the end of the administration of the probiotic gelatin or of the placebo. Several clinical assays using single probiotic strains or combinations of them administered to treat *H. pylori* infection differed in their results or showed no effect [59,60,61], while others showed eradication of the infection in some patients [62,63,64]. It has also been observed that the treatment with probiotics alleviates the symptoms of *H. pylori*-caused gastritis and reduces the colonization by this pathogenic bacterium. As might be expected, thus far, there are no reports of the complete eradication of *H. pylori* infection by the administration of a single probiotic [65]. Nevertheless, it has been demonstrated that complementary treatments with probiotics benefit infected persons by decreasing the harmful effects of antibiotics used to eradicate *H. pylori* [8]. The long-term administration of probiotics might have positive results in *H. pylori* infection, particularly in reducing the risk to develop diseases caused by the high levels of gastric inflammation [65]. Moreover, the results of the present work confirm previous results reported for this probiotic strain [21,22,26,50,66,67], which may even be considered as a potential immunobiotic capable of reducing the adhesion of *H. pylori* to the gastric mucosa and the inflammatory damage caused by this pathogen, modulating the immune responses at the intestinal mucosa and systemic levels [22].

## 5. Conclusions

In conclusion, this clinical assay confirmed the prophylactic effect of the probiotic strain *L. fermentum* UCO-979C, administered in gelatin, against infection by *H. pylori,* having a 92.6% efficacy in avoiding infection by this pathogen in non-infected individuals. Additionally, it allowed us to collect, for the first time, important data about our region on *H. pylori* infection in the 18–30-year-old population, which showed a 42.7% prevalence.

## 6. Patents

The invention patent was granted by INAPI, Chile. “*Lactobacillus Fermentum* Rgm 2341 Strain and Its Use in the Prevention and Treatment of *Helicobacter pylori* Infection”. Registration number: 63992.

## Figures and Tables

**Figure 1 foods-11-01668-f001:**
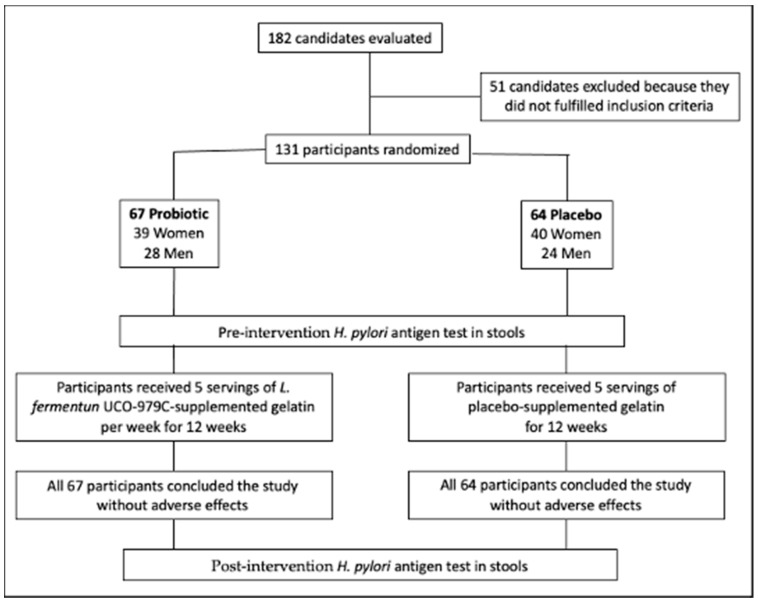
CONSORT flow diagram of recruitment and analysis of participants.

**Figure 2 foods-11-01668-f002:**
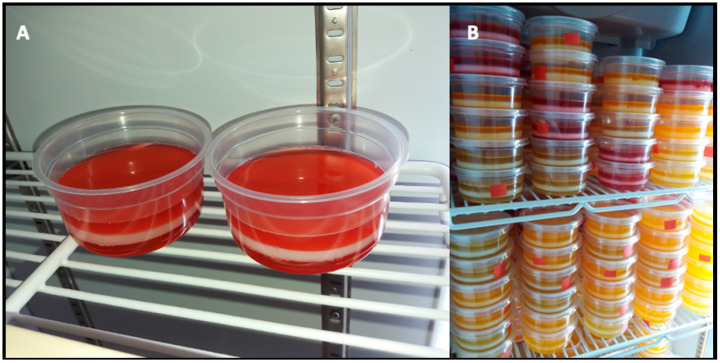
(**A**) Three-layer probiotic-supplemented gelatin (**right**) and three-layer placebo-supplemented gelatin (**left**), the probiotic was included in the middle layer; (**B**) refrigerated gelatins in storage to be distributed to the participants.

**Figure 3 foods-11-01668-f003:**
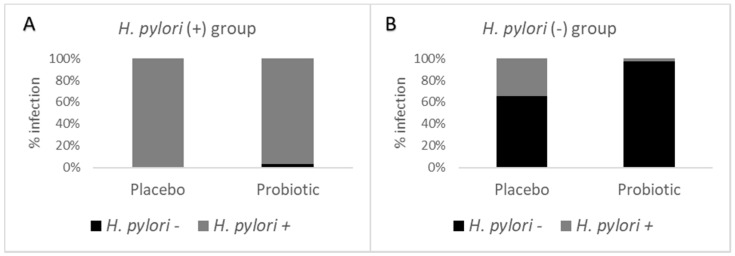
Effect of consuming the probiotic gelatin (100 g serving) containing 10^7^ CFU mL^−1^ *L. fermentum* UCO-979C strain (in 100 g serving) for 12 weeks on the percentage of *H. pylori*-infected participants at the end of the intervention. (**A**) *H. pylori*-infected participants before the intervention; (**B**) *H. pylori* non-infected participants after the intervention. *H. pylori* (−): *H. pylori* non-infected participants; *H. pylori* (+): *H. pylori*-infected participants.

**Table 1 foods-11-01668-t001:** Demographic and clinical characteristics of participants in the groups receiving probiotic- or placebo-supplemented gelatin.

Demographic and Clinical Characteristics	Placebo (N 64)	Probiotic (N 67)	*p*-Value
Age in years (average ± DS)	22.34 ± 2.92	22.67 ± 2.84	0.3274
Gender			0.7213
Women	40	39	
Men	24	28	
Pathology			
Irritable bowel	4	2	0.4399
Diabetes	3	1	0.3650
Lupus	0	1	>0.9999
Gastroesophageal refluxs	0	1	>0.9999
Gastritis	0	2	0.4960
Lactose intolerance	8	4	0.2354
Alcohol consumption	46	39	0.1426
Asthma	2	6	0.2732
Allergy	3	5	0.7176

**Table 2 foods-11-01668-t002:** *H. pylori*-infected and non-infected participants before and after the end of the treatment.

Intervention	Pre-Treatment	Post-Treatment		*p*-Value
		*H. pylori* (+)/%	*H. pylori* (−)/%	
*H. pylori* (+)				>0.9999
Placebo	26	26/100	0/0	
Probiotic	30	29/96.7	1/3.3	
Total	56	55/98.2	1/1.8	
*H. pylori* (−)				0.0005
Placebo	38	13/34.2	25/65.8	
Probiotic	37	1/2.7	36/97.3	
Total	75	14/18.7	61/81.3	

*H. pylori* (+): *H. pylori*-infected participants; *H. pylori* (−): *H. pylori* non-infected participants.

## Data Availability

Not applicable.

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
