# Peer review of "Consumption of a Gelatin Supplemented with the Probiotic Strain Limosilactobacillus fermentum UCO-979C Prevents Helicobacter pylori Infection in a Young Adult Population Achieved"

_foods, 2022, doi:10.3390/foods11121668_

Round 1

Reviewer 1 Report

This manuscript evaluated the impact of administering a gelatin supplemented with the probiotic strain L. fermentum UCO-979C on the H. pylori infection in asymptomatic Chilean young adults. It is well written. The clinical studies of the probiotics are of interest and the findings should provide prove for the use of the studied probiotic. Some minor comments are as follows:

-Check the format and spellings carefully e.g. “infection” in the title should be italic. Line 229, check scientific name which should be italic. Please carefully check throughout the manuscript.

-Introduction should provide more information about the gelatin being used as the carrier for the probiotic. Why it is being used? The authors mentioned about “vegetarian” but gelatin is not the vegetarian food (Line 63). Any previous studies that support the survival of probiotic in gelatin?

-Methods for clinical trials seem complete. EC approval information was also supplied in the manuscript.  

-Results: Though this manuscript focused on clinical studies but it should be better if the authors provide the detailed information about the survival of probiotic in the gelatin. How can the authors be so sure the stability of the gelatin supplemented with the probiotic over a week period?

-Discussion seems complete and extensive.

Author Response

Dear Reviewer, we deeply appreciate all your comments which allowed us to significantly improve our manuscript and we will make sure to keep them in mind in our future work. The new line numbers make reference to the version with control of changes. 

Thank you very much.

The authors

Comments and Suggestions for Authors

This manuscript evaluated the impact of administering a gelatin supplemented with the probiotic strain L. fermentum UCO-979C on the H. pylori infection in asymptomatic Chilean young adults. It is well written. The clinical studies of the probiotics are of interest and the findings should provide prove for the use of the studied probiotic. Some minor comments are as follows:

-Check the format and spellings carefully e.g. “infection” in the title should be italic. Line 229, check scientific name which should be italic. Please carefully check throughout the manuscript.

Sorry for the typo, “infection” is not in italics anymore (line 2).  Regarding the scientific name in line 229 it was H. pylori (now line 499).

-Introduction should provide more information about the gelatin being used as the carrier for the probiotic. Why it is being used? The authors mentioned about “vegetarian” but gelatin is not the vegetarian food (Line 63). Any previous studies that support the survival of probiotic in gelatin?

Thank you for your comment.  We cannot but agree with you that including a mention to vegetarians was a big mistake.  Sorry for that. 

Regarding the use of gelatin, reasons to use it in the food industry and reports on the benefit of gelatin as the supporting matrix for probiotics were included in the Introduction (lines 220 to 227).

-Methods for clinical trials seem complete. EC approval information was also supplied in the manuscript.

In accordance with the comment of the Reviewer, we understood that the section Materials and Methods does not require modifications.

-Results: Though this manuscript focused on clinical studies but it should be better if the authors provide the detailed information about the survival of probiotic in the gelatin. How can the authors be so sure the stability of the gelatin supplemented with the probiotic over a week period?

Sorry we did not include this information in the manuscript. It is correct that gelatins were prepared and distributed to the participants in a weekly basis.  But random servings of gelatin were seeded weekly for four weeks on MRS medium to count viable L. fermentum UCO-979C.  Results indicated that L. fermentum UCO-979C counts were >107 UFC ml-1 for three weeks and it decreased by 1 logarithm after the fourth week.  The procedure is now included in Materials and Methods, at the end of Section 2.4 (lines 330 to 336) and the result in Results, Section 3.3 (lines 453 to 458).  Also, this issue is now included in the Discussion (lines 615 to 621).  It may be worth mentioning that L. fermentum UCO-979C strain is capable to maintaining a viability above 108 CFU/g during the 90 days of storage at -18°C in ice creams (Paucar-Carrion et al. 2022).

Paucar-Carrión, C. Espinoza-Monje, M., Gutiérrez-Zamorano, C., Sánchez-Alonzo, K., Carvajal, R. I., Rogel-Castillo, C., Saez-Carrillo, K., García-Cancino, A. 2022.  Incorporation of Limosilactobacillus fermentum UCO-979C with anti-Helicobacter pylori and immunomodulatory activities in various ice cream bases. 

- Discussion seems complete and extensive.

In accordance with the comment of the Reviewer, we understood that the section Discussion does not require modifications.

Reviewer 2 Report

The study addressed relevant issue in the field and it would be more of interest to the audience, however, there are too many syntax errors which are almost completely masked the intending meaning of most points being addressed by the author. This has to be improved for clarity. 

Line 13, change the sentence to “Helicobacter pylori is a bacterium associated with various…”

Line 21, remove either the comma or bracket from “…study, (56 participants),…”

Line 31, do authors mean host and stomach serve the same purpose in this expression “…the host/stomach wall”?

Line 163, what parameters were evaluated?

Line 191, “None of the 131 recruited participants, 67 receiving the probiotic and 64 receiving the placebo”. The sentence in the quotation is not clear

Author Response

Dear Reviewer, we deeply appreciate all your comments which allowed us to significantly improve our manuscript and we will make sure to keep them in mind in our future work. The new line numbers make reference to the version with control of changes. 

Thank you very much.

The authors

Comments and Suggestions for Authors

 The study addressed relevant issue in the field and it would be more of interest to the audience, however, there are too many syntax errors which are almost completely masked the intending meaning of most points being addressed by the author. This has to be improved for clarity.

The manuscript was fully reviewed, looking for errors and correcting them.

- Line 13, change the sentence to “Helicobacter pylori is a bacterium associated with various…”

Modified as requested by the reviewer, “associated to” was replaced by “associated with” (line 15).

- Line 21, remove either the comma or bracket from “…study, (56 participants),…”

Sorry about the typo, removed the comma (line 23) and changed the values to percentages

- Line 31, do authors mean host and stomach serve the same purpose in this expression “…the host/stomach wall”?

Sorry for the typo.  The sentence was rewritten and the new version is “Helicobacter pylori is a Gram-negative bacterium having the ability to colonize the stomach wall and other structures of the host” (lines 34 to 35). 

- Line 163, what parameters were evaluated?

As requested in your comment, the parameters evaluated in each one of the participants are now listed in Materials and Method, Section 2.5 Data collection (lines 343 to 348).  The results were already included in Table 1 (Section 3.2 of Results) of the uploaded manuscript.

- Line 191, “None of the 131 recruited participants, 67 receiving the probiotic and 64 receiving the placebo”. The sentence in the quotation is not clear

Sorry for the typo.  The sentence alluded by you was rewritten.  It now reads: “Among the 131 recruited participants, 67 received the probiotic and 64 received the placebo.” (lines 438 to 439).

Round 2

Reviewer 2 Report

Much improved.